# A Robust Localization System for Inspection Robots in Sewer Networks [note 1]

**DOI:** 10.3390/s19224946

**Published:** 2019-11-13

**Authors:** David Alejo, Fernando Caballero, Luis Merino

**Affiliations:** 1School of Engineering, Universidad Pablo de Olavide, 41012 Sevilla, Spain; daletei@upo.es; 2Department of Systems Engineering and Automation, Universidad de Sevilla, 41009 Sevilla, Spain; fcaballero@us.es

**Keywords:** localization, sewer network, field robotics, Monte Carlo Localization, GPS-denied, underground robotics, global pose estimation

## Abstract

Sewers represent a very important infrastructure of cities whose state should be monitored periodically. However, the length of such infrastructure prevents sensor networks from being applicable. In this paper, we present a mobile platform (SIAR) designed to inspect the sewer network. It is capable of sensing gas concentrations and detecting failures in the network such as cracks and holes in the floor and walls or zones were the water is not flowing. These alarms should be precisely geo-localized to allow the operators performing the required correcting measures. To this end, this paper presents a robust localization system for global pose estimation on sewers. It makes use of prior information of the sewer network, including its topology, the different cross sections traversed and the position of some elements such as manholes. The system is based on a Monte Carlo Localization system that fuses wheel and RGB-D odometry for the prediction stage. The update step takes into account the sewer network topology for discarding wrong hypotheses. Additionally, the localization is further refined with novel updating steps proposed in this paper which are activated whenever a discrete element in the sewer network is detected or the relative orientation of the robot over the sewer gallery could be estimated. Each part of the system has been validated with real data obtained from the sewers of Barcelona. The whole system is able to obtain median localization errors in the order of one meter in all cases. Finally, the paper also includes comparisons with state-of-the-art Simultaneous Localization and Mapping (SLAM) systems that demonstrate the convenience of the approach.

## 1. Introduction

Many modern cities would not be as we know them without their sewer networks, which are vital for ensuring a healthy environment. However, the state of this infrastructure has to be continuously assessed in many different ways to ensure that it works properly, and to determine whether there are health risks in it or not. This is a harsh task in which robotics and automation could provide many benefits. For instance, consider the sewer network of the city of Barcelona in Spain, in which the experiments of this paper are carried out. The network has a total length of approximately 1532 km, in which more than 50% should be directly visited by operators. For them, sewers are unhealthy places that also present risky elements such as slippery floors, obstacles or biological hazard from the potential contact with waste water. Therefore, it would be very convenient to deploy a mobile sensing unit into the sewer network that can raise alarms if damages, blockages and other environmental hazards are discovered, as is proposed in the Echord++ PDTI for Urban Robotics Challenge [1]. According to operators of sewer inspection, these alarms need to be localized in global geographical coordinates, and with an accuracy of tens of centimeters. This is necessary in order to make the operators able to identify the places in which maintenance work should be carried out.

In this paper, we consider a mobile robot for this task. Note that a mobile robot unit is preferable to a sensor network because the length of the sewer network makes the latter unpractical. Developing an autonomous sewer inspection robot presents a wide variety of challenges: the environment is very aggressive with a high level of humidity and toxic gases; the robot should move in very narrow spaces with almost no place for maneuvering; communications are very restricted due to lack of direct line of sight; and there is no coverage of global localization systems, like GPS, to allow the robot to know its position. In addition, the robot needs to carry a wide range of sensors not only for self-localization and navigation, but also to perform the inspections tasks: cameras for visual imaging with artificial illumination, or 3D scanning sensors for metric environment reconstruction to facilitate impairment detection or devices for air/waste sampling. These sensors increase the weight of the system and also drain the power of the robot with the corresponding impact in battery life and autonomy, which is specially noticeable when using aerial robots. The system considered in the paper is a mobile ground robot (see Figure 1) equipped with several RGB-D cameras that make the robot able to identify and localize defects within the sewer network.

The paper focuses on the problem of robot global localization in sewers; that is, the problem of localizing in geographical coordinates a robot navigating through the sewer network. This problem is specially complex not only due to the lack of coverage of any Global Navigation Satellite System (GNSS) signals, but also due to the inherent symmetry of such places, which prevents constraining the position of the robot along the tunnel axis, the frequent slippage of wheels which render wheel odometry very uncertain, and the lack of features in certain cases, which also makes difficult to apply LIDAR and/or visual-based odometry.

The paper presents a novel system for global position tracking of a ground mobile robot in a sewer network. The system combines visual odometry (VO) and wheel odometry for estimating the motion of the robotic platform in a robust way. To correct for accumulating odometry errors and achieve global localization, the system employs a topological map with certain metric information from the Global Information System (GIS) used by the sewer operators, and a machine learning based manhole detector able to recognize the manholes using depth images. Geometric features from the sewer network are also used to further correct the estimation.

This paper is based on our previous work [2], which is largely extended in several important aspects, which can be summarized in the following points:The complete final system envisaged for the localization of a ground vehicle navigating in a sewer network is presented, which considers the following new elements:
–We have implemented a method for fusing visual and wheel odometry that increases the robustness of the localization system.–We have introduced a new method for estimating the relative orientation of the robot w.r.t. the sewer gallery.–The Convolutional Neural Network(CNN)-based manhole detector has been improved and trained with data acquired from different places.The experimental validation of the method has been greatly extended. We tested it in experiments carried out in different parts of Barcelona.We include comparison with different state of the art SLAM techniques.

The paper is organized as follows. Next section summarizes related work on localization, focusing in localization in confined spaces such as sewers and mines, to name a few. Section 3 overviews the main components of the robotic system used for sewer inspection. Section 4 describes the use of GIS in this paper, while Section 5 describes the localization method. Section 6 describes the odometry approach used, which fuses information from two sources. Section 7 and Section 8 describe the machine learning method for manhole detection and the relative orientation estimator of the sewer galleries, respectively. Then, Section 9 discusses the experimental results obtained in real sewers. Finally, Section 10 summarizes the main conclusions of this paper.

## 2. Related Work

The localization of a robot in a sewer system has a number of issues to be considered. First of all, most systems rely on odometry information from wheel encoders and inertial units. However, humidity, water and waste significantly decrease the wheels’ grip, distorting the computed linear and angular velocities based on wheel encoders. Besides, the robot has often to negotiate steps in order to cross wastewater channels, and in this process the robot can lose grip, further distorting the wheel odometry to a great extent. Even though some devices can improve the odometry reliability [3], these errors cannot be ignored in the long term.

One of the first localization systems, for the KURT inspection robot [4], already recognizes the main problems for odometry, like wheel slippage. Topological localization is done by recognizing junctions from data and matching them to a map. Wheel odometry is considered in [5], where also junctions are classified using readings from wheels and used to compensate the errors. Only simulations are considered.

LIDAR-based systems are also considered for motion estimation in tunnel-like environments [6]. The problem in this case is the lack of features and the symmetry of the environment, which makes it difficult to estimate certain degrees of freedom. Tethered robots can use also the tether to estimate the motion of the robot [7]. However, the use of tethers in the system presented in this paper was not an option because of the requirements of the local authorities. The same paper considers also VO for robot motion estimation. The visual odometer is based on the known geometry of a cylindrical sewer. In contrast, we consider a model-free VO approach, as the sections of the sewer are not uniform and their geometry is not always precisely known in advance. Also, in [8] the fading period of RF signals in tunnels (assimilated as waveguides) are used to obtain an odometry-like measurement for localization.

Besides odometry estimation, many inspection applications in sewers require global localization. Some systems are based on the communication of extra-low frequency signal with an external mobile system [9] to estimate the position of the robot. However, this requires additional infrastructure and cannot be applied to all types of sewers. A localization system is also presented in [7], based on matching the same visual features used for VO stored in a map. However, results are obtained in a small sewer testbed. Also the work in [10] proposes a similar localization system for 3D localization in sewer systems. Preliminary experimental results are shown, but also in small sections of the sewer system. In addition, 3D localization is not required in our application.

More recently, a localization system used for robots inspecting mines has been developed in [11]. Similar to ours, this work reduces the localization problem from the naive problem of determining the sequence of 6D poses of the platform to a reduced representation of 2D poses due to the topology of the scenario (see Section 5). The results seem promising as the proposed system is able to localize the robot with errors of less than 10 m. However, the performance attained does not meet our requirements. Moreover, the system should be supervised by an operator to be able to reach the aforementioned performance. This work has been significantly improved in [12], where the movement of the robot in 2D is estimated with a camera pointing to the ceiling, obtaining real-time localization in mines of few hundreds of meters long with a mean accuracy of 1.2
m. However, the environment should be mapped with precision and the images need to be labeled beforehand to use the proposed method, preventing its application in unmapped regions. This limits the applicability of the method in the sewer network of Barcelona, due to its length.

Reference [13] presents another interesting localization method for aerial platforms in GPS-denied spaces where radiation sources should be identified and their position should be estimated. They use the visual inertial odometry system proposed in [14] to localize the platform in 3D and to generate a local 3D map. This localization system has acceptable results given the limited flight endurance of the platform and estimates the position of the radiation sources by trilaterating radiation measures and actively searching for them. However, the localization method cannot be used in sewer networks because the length of the missions in the case of our platform, up to four hours, would make the odometry estimation diverge. To improve the long-term performance of the VO, some methods generate key-frames that are then related by a graph. Then, loop-closure detection can be used for generating congruent graphs and thus reducing the divergence of VO, as in [15,16]. Ref [17] extends the work of [15] by proposing a centralized structure of collaborative 3D map building in a team of multiple aerial vehicles. Nevertheless, online 3D map generation is not necessary in our system and the problem can be reduced to estimating the 2D pose of the platform in our environment. Moreover, we need a method that is able to estimate the global position of the robot.

Therefore, in order to make the system able to localize while inspecting more than 1 km per operation and make the system ready to be deployed in any location of the more than 800 km of visitable sewers in Barcelona [1], we make use of a graph obtained from GIS. This graph contains the global positions of elements of the sewer network, like manholes and forks. We thus propose the use of a Monte Carlo localization method in which odometry errors are compensated by recognizing those elements and by estimating the relative orientation between the robot and the current sewer section. To perform the aforementioned recognition, we employ machine learning techniques that have been proven very efficient to perform a great variety of complex tasks including speech [18], text [19], emotion [20] and activity recognition [21]; and traffic sign segmentation and identification [22].

## 3. Robotic System for Sewer Inspection

A new ground robot, the SIAR platform [23], has been developed to tackle the requirements of the sewer inspection application. It is a ground robot able to navigate through the sewage system with minimal human intervention and with the possibility of manually controlling the vehicle or the sensor payload whenever required. Figure 1 shows the robotic platform.

The robotic frame is made with sprayed water-proof encapsulations in order to accommodate for the hardest environmental conditions during sewer inspection. The robot has six wheels with independent traction on each wheel. It is able to navigate over a wide range of floors and small obstacles, including steps of over 20 cm. Interestingly, this platform is able to change its morphology thanks to a linear motor located at the bottom. This allows it to not only to navigate inside sewer galleries with different sections by changing its width but also to change the position of its center of gravity, increasing its maneuverability when negotiating steps and forks. This configuration is very flexible and adapts very well to the different scenarios the robot can find during routine inspections (see Figure 1).

The robot is equipped with a set of navigation, localization and environmental perception sensors. The main element of the sensor payload is a set of seven Orbbec Astra RGB-D cameras [24] that provides visual and depth information. There are three RGB-D cameras looking forward, three backwards and one upwards. For each direction, the robot has a camera parallel to the ground and two cameras facing downwards and sideways to visualize closer range obstacles. The upward-facing camera is placed over the center of the robot. This camera is used for detecting flaws in the tunnel dome and for identifying sewer elements such as manholes.

The robot also uses encoders to control the velocity of the traction motors and an encoder plus a potentiometer to control the position of its width actuator. The robot is also equipped with an inertial sensor. For more details on the SIAR robotic platform, the reader is referred to [25].

## 4. Using GIS Information

As specified in [1], the robot platform should be capable of navigate through 800 km of visitable sewers in Barcelona. It is not possible to map beforehand the whole sewer network with the robot. Thus, it is needed to use the GIS employed by the local sewer agencies, extracting the relevant information for localization. This information includes precise absolute (GPS) localization of the sewer galleries, the forks that should be traversed and the position of the manholes with errors that are usually of tens of centimeters. Moreover, there is usually information about the shape of the sewer gallery. All of this information is used by our localization system in the following ways:The localization of the robot should be confined to an area close enough to the sewer graph.Whenever a manhole is detected, the localization of the robot should be close enough to one of them.We add constraints to the orientation of the platform whenever the relative angle of the platform w.r.t. the current sewer gallery is estimated.

To automatically use the available information, we developed a utility that reads a DXF (Autocad) file provided by the local authorities in this case and generates a local graph in a desired area. Figure 2 represents an example of DXF input file and its associated sewer graph.

Figure 3 represents a detail of the graph used in our localization method. We can distinguish three main elements of the graph:**Sewer galleries**. The edges of the graph, in white, represent a sewer gallery that can be crossed by our platform.**Manholes**. The vertices of the graph that are represented in green squares represent a manhole in the sewer network. They are used to reduce the uncertainty of the localization of the platform.**Forks**. The edges of the graph marked in red squares represent a part of the network were two galleries that go in different directions meet. These vertices are called forks.

## 5. Monte Carlo Localization

The problem of localizing an object in the space consists of determining the 3D pose of the object with respect to some fixed reference frame. This pose can be represented with the following state vector, where the orientation is represented with Euler angles: $x={\left[x\;y\;z\;\gamma \;\psi \;\theta \right]}^{T}$, where x is the state vector of the robot, *x*
*y* and *z* are the 3D position of the robot in cartesian coordinates and the orientation is given in Roll-Pitch-Yaw standard, which are represented by γ, ψ and θ, respectively.

However, in this paper we assume that the roll and pitch angles are directly observable thanks to an onboard IMU. Moreover, the altitude of the robot can be estimated from the GIS information if *x* and *y* are known, in a similar fashion to the altitude estimation proposed in [26]. Thus, in this paper we should solve the problem of 2D global localization of the SIAR platform inside the sewers from local odometry and a simplified description of the sewer network. In this particular case the state vector to be estimated is:(1)$$l=\left[\begin{array}{c} x\\ y\\ \theta \end{array}\right]$$

The proposed approach is based on Monte Carlo Localization [27], which is a Markov Localization system that makes use of a particle filter to represent the robot localization belief. Hence, we define the distribution Bel(l) that expresses the robot’s belief for being at position l. To estimate this distribution, we use a set of particles where each one represents a hypothesis $l^{i}$, where *i* is the number of the particle.

There are three main stages in any Markov Localization system: initialization, prediction and update. We detail the particularities of each phase below.

### 5.1. Initialization

In the initialization, we set the initial Believe following a Normal distribution in the surroundings of the manhole in which the robot was deployed, setting the orientation of each particle according to the direction of exploration. In some cases, the position of the robot was not exactly below a manhole at the start of a mission. In these cases, we raised the variance of the normal distribution to take into account the uncertainty in the starting positions.

### 5.2. Prediction

In the prediction stage, the pose of each particle is updated taking into account the odometry measurements (see Section 6). We estimate the movement of each particle by using the following formulas:(2)$$\begin{array}{cc} x_{n+1}^{i} & =x_{n}^{i}+\left(\Delta r+\psi \right)cos\left(\theta_{n}^{i}\right) \end{array}$$(3)$$\begin{array}{cc} y_{n+1}^{i} & =y_{n}^{i}+\left(\Delta r+\psi \right)sin\left(\theta_{n}^{i}\right) \end{array}$$(4)$$\begin{array}{cc} \theta_{n+1}^{i} & =\theta_{n}^{i}+\Delta \theta +\xi \end{array}$$
where Δr and Δθ are the translational and angular displacements of the platform from the previous update state as measured by the odometry module, respectively; $\psi ∼N\left(0,\sigma_{\psi }\sqrt{\Delta x^{2}+\Delta y^{2}}\right)$ is the odometry noise in the translation and $\xi ∼N\left(0,\sigma_{\theta }\Delta_{\theta }+\sigma_{\theta }^{min}\right)$ is the odometry noise in the orientation. Note that we added a minimum variance to the latter distribution for taking into account cases in which Δθ is close to zero, which happened when navigating in long straight corridors. Our system can be fed with different odometry sources, including wheel odometry, VO and a more robust odometry that fuses the information from both sources, please refer to Section 6.

### 5.3. Update

The filter needs a method to validate that particles are in a consistent position taking into account the perception measurements. In this paper, we introduce three different validation processes that weight the particles in different ways depending on the information available from the sensors. This weight is used for calculating the mean position of the particle set. For example, whenever the manhole detector gives us a positive detection we rank the particles taking into account the distance to the closest manhole.

Next we detail the different ways of ranking the particles used in this paper.

#### 5.3.1. Edge Weighting: Measuring Lateral Errors

In the most common update case of the proposed method, the particles are ranked according to their distance to the closest edge of the graph extracted from GIS data. Particularly, we use Equation (5) to calculate the weight of each particle:(5)$$w_{edge}=\frac{1}{\sigma_{e}\sqrt{2\pi }}e^{-\frac{d_{e}^{2}}{\sigma_{e}^{2}}}$$
where $d_{e}$ is the distance from the particle to the closest edge and $\sigma_{e}$ is related to the width of the sewer.

It is worth to mention that the map could not be as accurate as desired and thus it could be a good idea to overestimate the edge deviation. This is more evident in the surroundings of forks and turns. Therefore, in this paper we use two different values of the standard deviation depending on the proximity of a fork. Whenever a fork is nearby (closer than a given distance) it is set to 0.6
m, otherwise is set to 0.3
m.

#### 5.3.2. Manhole Weighting: Determining the Longitudinal Position

In the case of navigating in long sewer corridors, which is very common in practice, it is difficult to obtain references in the advancing direction of the corridor. While with the previous weighting method we were able to force particles follow the network topology, and thus we can easily estimate which corridor are we traversing, it is not able in general to decrease the uncertainty of the localization estimation in the direction of the corridor (longitudinal uncertainty).

For this reason we make use of a manhole detector that let us known if we have reached this determinate point in the corridor. This way we can reduce the longitudinal uncertainty.

Therefore, we use Equation (6) to rank the particles of the filter whenever a manhole is detected by the manhole detection module.
(6)$$w_{manhole}=\frac{1}{\sigma_{m}\sqrt{2\pi }}e^{-\frac{d_{m}^{2}}{\sigma_{m}^{2}}}+w_{manhole}^{min}$$
where $d_{m}$ is the distance from the particle to the closest manhole in the graph and $\sigma_{m}$ is the considered standard deviation surrounding the manhole, which is related to the size of the area where they can be detected. The additional term $w_{manhole}^{min}$ term has been added to the weight in order to mitigate the effects of a false detection in places far away of manholes. We make its value follow Equation (7) for the particles in the filter:(7)$$w_{manhole}^{min}>>\frac{1}{\sigma_{m}\sqrt{2\pi }}e^{-\frac{d_{m}^{2}}{\sigma_{m}^{2}}}$$

Therefore, all particles are uniformly weighted in this case. On the other hand, in the surroundings of the manhole the most important term should be due to the normal distribution:(8)$$w_{manhole}^{min}<<\frac{1}{\sigma_{m}\sqrt{2\pi }}e^{-\frac{d_{m}^{2}}{\sigma_{e}^{2}}}$$

This equation must hold if $d_{m}<d_{detect}$, where $d_{detect}$ is the maximum distance in which we can detect a manhole.

#### 5.3.3. Angular Weighting

The last way to rank the set of particles takes into account an estimation of the relative angle between the robot and the gallery it is going through, which is estimated by the relative angular detector (see Section 8). Let $\theta_{r}$ be the orientation of the robot, $\theta_{g}$ the orientation of the gallery, as given by the GIS, and Δθ the measure of the relative orientation of the robot with respect to the gallery. We use Equation (9) to rank the particles.
(9)$$w_{edge}=\frac{1}{\sigma_{a}\sqrt{2\pi }}e^{-\frac{{\left(\theta_{r}-\theta_{g}-\Delta \theta \right)}^{2}}{\sigma_{a}^{2}}}$$
where $\sigma_{e}$ is a term that reflects the uncertainties of both the measure and the GIS.

It is worth to mention that we do not apply this kind of weighting any time that a measure is available. Instead, when the detector is giving measures, we alternate the angular and edge weighting because we should also verify that the hypotheses follow the graph topology. Moreover, we do not use the angular measurements when an hypothesis is close to a fork. In these cases, the use of this update can distort the shape of the set of particles as the nearest edge related to each particle can vary and thus two particles that are close in distance could have very different relative angles.

### 5.4. Resampling

Whenever the dispersion of the particle set exceeds a threshold, or the number of maximum number of updates is reached, the set of particles is restructured through importance resampling. A new set of particles is obtained by randomly sampling the old set with a sampling distribution proportional to the weights of the particles. We make use of the low variance sampler described in [28]. By periodically resampling the particles according to the weights proposed in the previous sections, we ensure that most of the particles are located according to the prior GIS information.

## 6. Robust Odometry

Robot odometry is one of the main elements into the localization architecture. The more robust and accurate the odometry is, the better is the robot localization in general. Figure 4 represents the different methods and steps used in this paper for obtaining odometry estimations. These sources include VO using the RGB-D onboard cameras, wheel odometry fused with IMU information, and a fused odometry from both sources.

### 6.1. Wheel Odometry

Ground robots use to rely on wheel encoders and inertial units to estimate its odometry, as this is a good sensor combination with proven good results in many scenarios. Assuming no slippage, we can easily estimate the translation in the direction where the platform is facing by multiplying the mean angular displacement of the wheels times its radius.

We could also use wheel encoders for estimating the rotation of the platform. However, in this case we take advantage of the onboard IMU, which gives us a better estimation of the rotation of the platform in the three axes. Nevertheless, magnetometers are not reliable enough in sewers due to the presence of ferromagnetic materials, so it cannot be used for BIAS compensation in Yaw angle. Instead, an Extended Kalman Filter (EKF) is used to fuse wheel encoders and gyroscope measurements in order to estimate the yaw BIAS.

However, the environmental conditions of the sewers like humidity, water and waste significantly decrease the wheel grip can distort the computed linear and angular velocities based on wheel encoders. It forces us to include more sensing modalities in order to have an accurate robot odometry. Moreover, the grip can be partially lost for short periods of time when executing complex maneuvers such as managing forks.

### 6.2. Visual Odometry

The visual odometry system proposed in this paper estimates the relative movement of the platform based on information from the onboard RGB-D cameras. The system also makes use of the onboard IMU for stabilizing the robot roll and pitch, as they are fully observable from the IMU accelerometer. The visual odometer used in this paper is an adaptation of previous author work presented in [29] and extended to RGB-D cameras (our RGB-D visual odometer is is available at [30]). Next paragraphs summarize the different stages in the VO computation.

#### 6.2.1. Image Feature Detection

With every camera frame, a set of interest points is found in the color image. These interest points are visual features extracted using the Features from Accelerated Segment Test (FAST) algorithm [31], which is a method for corner detection with a strong computational efficiency. The identified interest points contain local information that ideally make them repeatable across consecutive frames. We impose a minimum and maximum number of features to be extracted from equally divided regions of the images, commonly known as buckets. This allows us to distribute the features as most homogeneously as possible over the whole image, enhancing the numerical stability of the odometry. For each point, its 3D position is estimated thanks to the depth field of the RGB-D camera.

#### 6.2.2. Image Feature Description and Matching

The next step involves finding correspondences between the current image and previous one. A feature point description is needed in order to compare and find similarities between each pair, and for that the general-purpose Binary Robust Independent Elementary Features (BRIEF) descriptor [32] is used. This descriptor is very efficient since it is based on simple intensity difference tests, and the similarity evaluation is done using the Hamming distance. Together with the computational efficiency, BRIEF descriptors are especially interesting for feature tracking under small visual perturbations, a very usual situation in VO.

#### 6.2.3. PnP-based Relative Pose Estimation

Once a set of enough robust common features has been found between consecutive frames, this set of matches is used to solve a Perspective-n-Point problem (PnP), i.e. the pose estimation from n 3D-to-2D point correspondences (please recall that for each point we have its 3D position). This allows to estimate the camera rotation and translation that minimizes the re-projection error of the 3D points of previous frame into the points of the current frame. This is a non-linear optimization that can estimate the camera transform very accurately at a reasonable computational cost. In particular, we use the EPnP algorithm [33], which provides an efficient implementation of the pose change estimation between both images.

#### 6.2.4. Key-Framing

Finally, a key-framing approach has been adopted in order to partially mitigate the effect cumulative errors in odometry. Thus, new key-frames are produced only when the feature tracking flow exceeds a given threshold. At that moment, the current image descriptors, their 3D estimated position and the robot position/orientation are stored. We compare the descriptors of the subsequent frames with respect to ones of the last key-frame (and their pose), instead of the immediately preceding image. In this way, the errors are only accumulated with the introduction of new key-frames.

### 6.3. Fusing Odometry

Each odometry system proposed in this paper has its advantages and drawbacks. On the one hand, wheel odometry is usually the most reliable to estimate the advance of the platform, provided that no significant amount of slippage exists and that the robot does not lose the traction. On the other hand, RGB-D VO can obtain a full 3D robust estimation with accuracy that can be used for obtaining local 3D maps in a determinate region. However, it can underestimate the translation displacement when there is a lack of matches between a frame and the previous key-frame. Thus, if the number of matches is below a threshold the results are discarded. Furthermore, the results are also discarded if the odometry estimate returns translational or rotational movements that exceed given thresholds. This is done because these circumstances usually reflect that there were too many mismatches. This is usually referred as a odometry jump.

For this reason, as a last step, we fuse the different odometry inputs: the estimation obtained by VO is compared to the one provided by the wheel encoders. We check that the relative difference between angular or translation displacements estimated by both methods does not go beyond a threshold during a determinate time window (in our experiments we set it to 75%). If the check fails, we suppose that the robot lost the grip and the system falls back to VO, provided that the VO did not fail during that time window (due to lack of matches or a odometry jump). Otherwise, we use the estimation from wheel odometry.

## 7. Manhole Detection

The high symmetry of the sewer gallery makes localization a very complex problem. Commonly used tools for localization as visual place recognition algorithms usually suffer from low accuracy when targeting repetitive visual structures [34]. This problem is accentuated by the poor illumination conditions and high symmetry of the environment. On the other hand, detailed 3D/2D maps [35,36] of the sewer network are frequently unavailable due to its vast size.

Manholes offer a good opportunity for localization. First, there exists a regulation of the maximum distance between two consecutive manholes for safety reasons. Second, their particular shape, their diameter (70 cm) and the fact that they break with the uniformity of the gallery ceiling enables their detection with an acceptable success rate (see Figure 5). Third, their positions are labeled in the Global Information System (GIS) provided by the local agencies. This information can be used to reset the localization errors if the robot is able to identify the manhole on top of it.

Thus, our objective is to automatically detect if the robot is lying under a manhole. We use the depth images provided by a camera placed on top of the robot, pointing toward the ceiling. Our approach makes use of machine learning approaches to train a classifier able to split the depth camera view between regular ceiling and manhole.

Considering the significance of these detections for the localization system, we impose the following two hard constraints: the accuracy must be very high to avoid false positive or false negative detections that might impact in the localization results, and second, the manhole detection must be light-weight from the computational point of view to not compromise the rest of the navigation task.

With all the previous constraints in mind, CNNs emerge as a convenient technique as they have been extensively used for image classification purposes in last years with great success. Their ability to extract hight level features from the images and to easily reduce the dimensionality of the problem are key factors for their success. In addition, this dimensionality reduction helps to decrease the computational requirements of the classifier.

This paper extends and improves the manhole detection system presented in [2]. The number of filters per convolutional layer have been significantly increased, also adding a new fully convoluational layer in the output. These modifications, together with a wider dataset, allows us to improve the detection accuracy from 96% to 98%. Next paragraphs provide details on the network and the evaluation.

### 7.1. Convolutional Neural Network Architecture

The CNN architecture presented in this paper extends and improved the architecture presented by the authors in [2], which is based in the AlexNet architecture [37]. This artificial neural network architecture efficiently extracts low and high level features from the input images. These features are specially interesting for classification tasks, as pointed out in [38] and more recently in [39]. Thus, this architecture has been successfully exploited to build accurate image classifiers, as in [40]. However, these good detection capabilities come with the drawback of high computational requirements. Many different approaches have been proposed in the state of the art to decrease the computational requirements by means of input dimensional reduction [41,42]. The proposed approach follows these main guidelines, but it makes use of a low dimensonality input (80×60 pixels) to avoid high computational requirements. It is based on the combination of convolutional neural layers that sequentially process the information provided by the previous layer. The use of max-pooling allows to select the strongest features estimated in each layer. The final fully connected layers implements the detection function based on the high level features extracted at the end of the convolutional section.

The CNN architecture adopted is presented in Figure 6. Notice how this approach differs from the one presented in [2] in the number of filters per convolutional layer, and also in the size and structure of the fully connected layers. Thus, the first step consist in down-sampling the depth image to a smaller scale, 80×60 particularly. The network is composed by four convolutional layers followed by ReLU activations and MaxPooling, except for the last one. The output of the fourth convolution layer is fed into a fully-connected layers of 150 and 50 neurons, with ReLU activations. Finally, the output layer is composed by a single fully-connected neuron with sigmoid activation functions. The result is a binary value that is true whenever a manhole is detected and false otherwise.

The original CNN architecture proposed in [2] was composed by 8000 parameters approximately. The one proposed in this paper is larger, but as point out in Section 7.4, it improves the results with a minor impact in the computational time. The size of the convolutional layers in the proposed approach is still small, and also the number of layers. This design is on purpose, so the number of parameters is around the 110,000. We can see how the number of parameters is multiplied by a factor of 13 approximately, but the network can be applied over an image in less than one millisecond by using a regular CPU, as can be seen in Section 9.5.

### 7.2. Dataset for Training

The real datasets captured by the SIAR platform in the sewer system of Barcelona were used for training and validation of the proposed artificial neural network. The data of four experiments were selected to extract positive and negative examples to train and validate the manhole detector.

In both datasets, training and validation, a manual labeling of the depth images with manholes is done. The positive set of the training dataset is expanded by introducing horizontal and vertical flips, and small translations and rotations.

As a whole, the dataset is composed by 150,000 depth images samples with a resolution of 80×60. Each sample integrates a label that indicates whether the image contains a manhole or not. From these samples, we use 45,000 to validate the CNN that are not included into training process.

### 7.3. Training

We implemented the proposed network using Python and the Keras library [43]. The learning algorithm used to train the networks is the Adaptive Moment Estimation (Adam), an algorithm for first-order gradient-based optimization. Reference [44] showed that Adam is more effective when compared to other stochastic first-order methods on MLP and CNN.

Each learning process takes 40 epochs with a batch size of 100 samples. We include a 20% dropout layer [45] after every convolutional or fully connected layer (except for the output) to prevent the computed neural networks from overfitting.

### 7.4. Validation

After the training process, we validate the model with a set of 45,000 samples not used in the training (test set). The obtained accuracy results show a 98.36% in the validation set. These results are very satisfactory, taking into account that different types of galleries visited in both datasets and that there were opened and closed manholes.

Table 1 summarizes the confusion matrix of the proposed CNN-based detector. For the sake of clarity, we also include the detection capabilities obtained with previous authors work [2] in Table 2. It can be seen that the detection capabilities have been significantly improved with respect to the previous approach. The true positive rate have been improved from 90% to the current 95%, while the false positive rate has also been improved accordingly from a 10% to a 5%. The negative (true and false) detection rates have been also slightly improved w.r.t. our previous implementation.

## 8. Estimation of the Relative Orientation to the Sewer

Figure 7 represents the most common cross sections of the visitable sewers in Barcelona [1]. As represented, the vast majority of them have two lateral walls that can be easily detected with the depth information of the front camera onboard the robot. In this paper, we use the information of the planes in order to estimate the orientation of the robot w.r.t. the sewer gallery. This is done by calculating the relative angle between the robot and each wall plane.

### 8.1. Fast Plane Detection

As stated before, we are interested in searching for planar regions in the space to detect the lateral walls of the sewer gallery. A plane in the space can be modeled as:(10)n·r=d
where n is its normal vector, r is a point in the plane and *d* is the distance from the origin to the plane. We use the region-growing based plane detector first introduced in [46] in order to detect them.

Let $\rho_{i}$ be a measure of a captured point cloud by the sensing device in the direction $m_{i}$. We assume that the $\rho_{i}∼N\left({\hat{\rho }}_{i},\sigma_{i}\right)$, where ${\hat{\rho }}_{i}$ is the true range and $\sigma_{i}$ is an upper bound of the standard deviation of the measure. $r_{i}=\rho_{i}m_{i}$ is the position of the measure in the optical frame of the device.

Under these assumptions, given a set of points $p\equiv \left\{p_{j}:j\in 1\ldots N\right\}$, we can estimate the normal of the best fit plane as the eigenvector associated with the minimum eigenvalue of matrix *M*:(11)$$M=\sum_{i=1}^{N}\left(r_{i}-r_{G}\right){\left(r_{i}-r_{G}\right)}^{T}$$
where $r_{G}=\frac{1}{N}\sum_{i=1}^{N}r_{i}$ is the gravity center of the set of points. Then, the distance of the plane can be retrieved from:(12)$$d^{*}=n\;\cdot \;r_{G}$$

Finally, we can estimate the covariance matrix of the parameters of the plane with the equations:(13)$$\begin{array}{cc} \Sigma & ={\left(H\right)}^{+} \end{array}$$
(14)$$\begin{array}{cc} H & =\frac{1}{\sigma^{2}}\sum_{i=1}^{N}\left[\begin{array}{cc} r_{i}r_{i}^{T} & -r_{i}\\ -r_{i} & 1 \end{array}\right] \end{array}$$
where the operator ${}^{+}$ refers to the Moore-Penrose pseudoinverse, as the Hessian is not of maximum rank: it has a zero eigenvalue with associated eigenvector n (see [46] for more details).

### 8.2. Orientation Estimation from Vertical Planes

The proposed method looks first for large and vertical planar patches in the depth image (see Figure 8) of the front camera of the robot by using the method proposed in the previous section.

Whenever two large vertical planes are found, they are used in order to estimate the relative angle of the walls with respect to the robot. This relative angle is used to rank the hypotheses according to the available GIS information.

Let $P\equiv \left\{n\;\cdot \;r=d\right\}$ be one of the wall planes. The direction of the gallery can be estimated by intersecting the plane with a ground plane, which is represented as $G\equiv \left\{k\;\cdot \;r=0\right\}$ in a gyro-stabilized frame centered in the base of the robot, where k is the unit vector along the *z* axis. The intersection of these planes is a line pointing in the direction of the sewer gallery. A vector l in this direction is perpendicular to the normals of the two planes. Therefore:(15)$$\begin{array}{cc} l & =n\times k \end{array}$$(16)$$\begin{array}{cc} \theta & =\arctan\left(\frac{l_{y}}{l_{x}}\right) \end{array}$$
where θ is the relative angle of the gallery with respect to the robot.

We can evaluate the goodness of the orientation estimation by using the covariance of the planes obtained in Section 8.1. In particular, the variance of θ can be estimated by the following formula.
(17)$$\sigma_{\theta }^{2}=J^{T}\Sigma_{xyz}J$$
where $J^{T}$ is the Jacobian of Equation (16) with respect to $l_{x}$, $l_{y}$ and $l_{z}$ that can be obtained by applying Equation (18); and $C_{xyz}$ is the covariance of the normal vector, which can be obtained from the covariance matrix Σ as expressed in Equation (20). These matrices can be obtained by using the following formulas.
(18)$$\begin{array}{cc} J & =\left[\begin{array}{c} -\frac{1}{y\left(\frac{x^{2}}{y^{2}}+1\right)}\\ \frac{x}{x^{2}+y^{2}}\\ 0 \end{array}\right] \end{array}$$
(19)$$\begin{array}{cc} \Sigma_{xyz} & =-{\left(H_{\hat{n}\hat{n}}-H_{\hat{n}d}H_{dd}^{-1}H_{\hat{n}d}^{T}\right)}^{+} \end{array}$$
(20)$$\begin{array}{cc} H & =\left[\begin{array}{cc} H_{\hat{n}\hat{n}} & H_{\hat{n}d}\\ H_{\hat{n}d}^{T} & H_{dd} \end{array}\right] \end{array}$$
where we use a block decomposition of the Hessian *H* calculated in Equation (13). For a proof of Equation (20) please refer to [47], Appendix B.

### 8.3. Experimental Validation

To test the goodness of the detector, we performed a set of experiments in a corridor at the University Pablo de Olavide facilities with an Orbbec Astra depth sensor. In the tests, we use our proposed wall detector (our implementation of the wall detector is available at [48]) for estimating the orientation of the camera w.r.t. the corridor. Each experiment starts with the sensor pointing towards the direction of the corridor. Then, we rotate it an approximate number of degrees with respect to the corridor in each direction.

Figure 9 represents the estimated angle of the camera w.r.t. the corridor. It also shows the reference angle were the sensor was placed for convenience. Please note that the sensor was placed manually to a discrete set of angles for testing purposes. Therefore, we have to wait for the angular measure to be stable before comparing it to the reference value.

These results show that the the detector is able to precisely detect the rotation with an relative error of less than 10% in all the studied cases, which range from 10 to 30 degrees. Therefore, the localization estimation can be used to increase the robustness of the localization algorithm. Moreover, the estimation error is below of 0.03 rad in all cases, which can be used as a conservative standard deviation of the estimation. However, in the experimental results presented in this paper, we raised the standard deviation to 0.06 rad in order to take into account possible errors in the GIS data regarding to the direction of the gallery.

## 9. Experiments

We tested the proposed localization system in a wide variety of scenarios with real field data. The platform was operated in semi-autonomous mode without the need of operators nearby, with the exception of the deploy and recover procedures and a manual recovery in the experiments of October 17th. In this section, we detail the results obtained in three different real sewer scenarios. The scenarios were all placed in Barcelona (Spain), at the locations of Mercat del Born, Creu de Pedralbes and Passeig Garcia-Faria, in order of appearance. Readers are encouraged to reproduce the results, as the code of the proposed localization system can be openly accessed on [48], where instructions to run the simulations can be found. The data gathered from the experiments of the Echord++ project is available at [49]. All the code has been developed under the Robotic Operating System (ROS) [50], it is distributed under the MIT license and it has been tested in its Indigo, Kinetic and Melodic ROS distributions.

In the following, we evaluate different aspects of the system, and perform comparisons with other approaches. It is difficult to verify the accuracy of the trajectory obtained because the sewers are a GPS-denied area. To obtain a partial ground truth, we labeled the frames of the upper camera when it was just below a manhole. In this way, we can compare the output of the localization system with the actual manhole position as it appears in the provided GIS.

Due to the stochastic nature of the presented localization algorithm, unless otherwise stated, the distribution of the error over 35 runs is represented in each case. In the following, to represent the distribution of the localization errors, we use the convenient box plot representation.

### 9.1. Effect of the Angle Detector

First, we analyze the the effect of including the estimation of the relative orientation of the robot w.r.t. to the sewer galleries. The results have been obtained from the data gathered in an experiment on October 11th, 2017, which had been carried out at the Mercat del Born area. Figure 10 represents the scenario with an example of localization results obtained by the proposed system.

Figure 11 shows the localization errors of the proposed method whenever the robot passed below a manhole with and without incorporating the orientation estimation. In comparison, the proposed method is able to reduce the median error in almost all cases. Moreover, the distribution of errors presents lower dispersion when using the estimation of the relative orientation within the sewer. In particular, there are no cases in which the localization error goes beyond 2 m in this scenario.

### 9.2. Different Odometry Comparison

We performed another field experiment in Mercat del Born scenario on October 17th, 2017. Please refer to Figure 12 to see an example of the localization results obtained in the experiment. In this section, we use the data gathered in these experiments to compare the localization results when using RGB-D VO, wheel odometry and the fused odometry system proposed in this paper. We made 35 runs of the algorithms in the same conditions, considering the error characteristics of each odometry estimation, in order to compare the behavior of the localization system when being fed with different odometry sources.

Figure 13 summarizes the results obtained by our localization module when the platform passes below the fourteen starting manholes and using the three different methods for odometry estimation. In this case, the results obtained when using VO start quite well in the first five manholes, but then its performance begins to degrade. In fact, from manhole Passeig 4 the localization loses the track of the platform due to bad odometry measures (see Figure 14). The reason is that in some parts of the sewer, the RBG-D odometry module does not detect enough features in the image and thus it cannot estimate the movement. Therefore, our localization module underestimates the displacements of the platform, leading to a complete lose of the track of the platform, which is produced when the platform passes below manhole Passeig 3.

On the contrary, wheel odometry tends to overestimate the displacements of the platform. This is produced when the platform loses the grip in several parts of the experiment. It is more evident when passing below manhole Fusina 3 for the second time (see the bottom plot of Figure 13). The reason is simple: just before passing below that manhole the platform fell to the gutter and had to be recovered manually, and thus the grip was lost during large periods of time when trying to recover the platform in teleoperation. This resulted in a increase of error when the platform passes below manhole Fusina 3, whose distribution has a median of over 3 m. However, this effect is not enough to make our localization module lose the track in this case.

Figure 15 depicts the distribution of the localization error when the platform passes below the thirteen remaining manholes. In comparison, the version with fused odometry obtains the better median error in the majority of the cases, with the exception of manholes Comercial 2 and 5 and Passatge 3, in which the results are very similar.

### 9.3. Comparison with SLAM Approaches

We also compare the performance of our proposed localization method with two popular state of the art SLAM approaches: gmapping [51] and ORBSLAM-2 [15]. In the case of the gmapping method, we used the front and rear cameras to generate two emulated lasers to feed it. In the case of the ORBSLAM-2 method, it was fed with the RGB-D images generated by the rear camera, as the platform moved backwards at first.

We made the comparison in a new scenario, as the previous experiments had a track length of more than 1 km and there are no loops in the trajectory that allow the methods to perform loop closures. In this case, we use experiments that were carried out in the Creu de Pedralbes Square, also in Barcelona. The experiments were carried out on the 12th of June, 2018, lasting forty-five minutes, approximately, and the platform traversed a two way track of roughly 200 m. Figure 16 represents the satellite view of the experiments at surface level overlaid by the localization results of each method.

As expected, the results of other methods diverge in the long-term, as not any external references are given to them. The main purpose of this section is to illustrate the difficulties of employing these methods in these kinds of environments.

In the case of the gmapping (blue trajectory in Figure 16), the generated map is also represented and, while it follows the shape of the real sewer, errors in the orientation of the gallery generate divergence in the results between the real gallery and the estimated by the module. Moreover, the algorithm usually duplicates some galleries as it does not estimate the displacement in the direction of the corridor with enough accuracy. In these cases, gmapping could not reconstruct the sewer topology faithfully.

Obtaining results from ORBSLAM-2 (purple trajectory in Figure 16) was more difficult as it lost the track several times during the experiment. Unfortunately, the original ORBSLAM library would not continue mapping until it regained the track. To overcome it, we made the algorithm ignore the lose of track and generate a new keyframe by supposing that no movement was produced. In this way, we ensure that the algorithm continues doing SLAM at the expense of introducing inaccuracies in the model. With this subtle modification we acquired a 3D map of the sewer in all the executions. In contrast to the gmapping results, we did not detect any duplication of galleries as this algorithm uses the RGB features and is able to track them back when going backwards.

Figure 17 represents the errors obtained by each method whenever the SIAR platform reached the manholes A or B (see Figure 16). The mean and standard deviation of ten executions of each method are represented. These results indicate that the gmapping is able to get more precise trajectories when compared with the ORBSLAM-2 method. However, our localization method clearly outperforms them.

### 9.4. Experiments in Passeig Garcia Faria: Determining Errors in GIS Data

Last, we describe the localization results of the final experiments carried out in the Passeig Garcia Faria. In this case, the robot followed a straight line gallery with a total distance of more roughly 400 m. Figure 18 represents a view of the Scenario in this experiment, as well as the output of our localization system. Figure 19a shows that the distribution of the localization errors whenever the robot passed below a manhole is always centered somewhere close to 1 m. However, there is one anomaly in this distribution: whenever the platform reached MH109, a systematic error in the localization which has a median of over 10 m was found. This fact suggests that the provided GIS data has a noticeable error with regards to MH109. Figure 19b represents the position of the Manhole as stated in the GIS and as estimated by our localization method.

### 9.5. Computation Time Analysis

In this section, we study the computation time of the main parts of the proposed localization system. As stated before, the code is implemented under ROS, which is designed to distribute the computation requirements of each module into different processes. This division is convenient, as it allows us to easily parallelize the execution of the different modules in the system. The results presented in this section have been obtained in a Intel^©^ Core^™^ i7-7700HQ CPU running at 2.80 GHz with 16 GB of RAM under Ubuntu 18.04 and ROS Melodic.

The main modules to be analyzed are the following:**Monte Carlo localization**. In this case, we measure the time spent in calculating the prediction and update stages of the filter, which are called whenever a threshold in rotation or translation with respect to the previous update is reached.**Orientation estimation**. Whenever a depth image from the front camera is received, the wall planes are detected and the orientation of the robot with respect to the gallery is calculated as seen in Section 8. We include the time spent in performing such estimation.**Manhole detector**. Whenever an image from the camera pointing to the ceiling is captured, we process it through the CNN described in Section 7 in order to determine whether the robot is under a manhole or not. We measure the time spent to process the image.**RGB-D Odometry**. Whenever an image from the front camera is captured, the RGB-D odometry module described in Section 6 estimates the displacement w.r.t. the previous image. This is the most time-consuming part of the fused odometry system proposed in this paper. We measure the time spent in the RGB-D odometry estimation.

Figure 20 represents the distribution of the execution time of the aforementioned processes gathered during the execution of the experiment described in Section 9.2. It shows that the least demanding process is the CNN query, which only needs an averaged time of 0.26 ms (see Table 3) to run. The remaining three processes have a similar mean execution time which ranges from few hundredths of a second to one tenth. This result allows us to use the RGB-D odometry and the orientation estimation modules at a frequency of 10 Hz. Table 3 also has information about the number of times each process is called during the experiment and its total execution time. Note that, even though the update of the particle filter has the highest unitary execution time, it is less time consuming in total because it is called roughly twenty times fewer when compared to RGB-D Odometry and to the orientation estimator. Finally, the proposed localization system as a whole used less than one CPU core during the whole experiment.

## 10. Conclusions

This paper has presented a system for ground robot localization in sewer networks. The system employs a robust odometry and GIS information in the form of manhole locations and to maintain a global localization of the robot in geographical coordinates. Using this, the inspection results obtained by the robot can be located and used by the operators to plan the required interventions.

The system has been intensively tested in real sewers, showing that it can be used to in operational scenarios, and bringing the required accuracy for day-to-day inspection operations. This is relevant to enhance the working conditions of operators in such environments.

One of the main conclusions of the study is the importance of a reliable odometry for good localization in such environments, characterized by frequent wheel slippage and often lack of visual features. A combination of wheel and RGB-D odometry is used here to overcome the drawbacks of both approaches.

The paper also shows how the metric information from the GIS, such as the position of sewer galleries, manholes and forks, can be used to successfully localize the robot, without requiring a full 3D detailed map of the environment. The current system employs a manhole detector for reducing the longitudinal uncertainty of the pose estimate. But other elements included in the GIS can be used to refine the localization, like inlets. The detector can be trained with different examples of such classes, and the system can be easily extended to consider them.

Future work includes, as mentioned, the consideration of additional elements in the localization process. Furthermore, the system can be extended to update and correct the information stored in the GIS, as it can sometimes have erroneous information. Devising ways to automatically detect such potential errors is a potential venue for future work. The ideas of the system can be also considered in other tunnel-like underground environments, which still remain a challenging scenario for robotics.

## Figures and Tables

**Figure 1 sensors-19-04946-f001:**
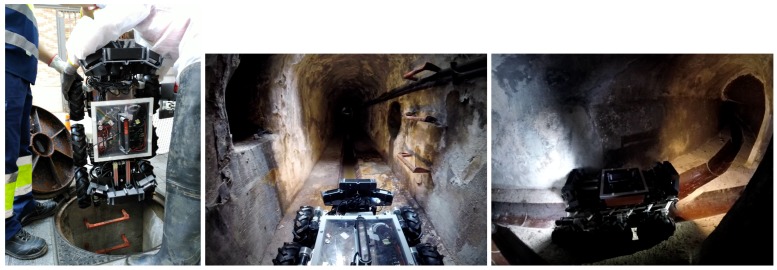
These pictures show the SIAR platform used in real experimentation. It is a six-wheeled ground robot. **Left**: being introduced through a manhole. **Center** and **right**: the typical environment considered.

**Figure 2 sensors-19-04946-f002:**
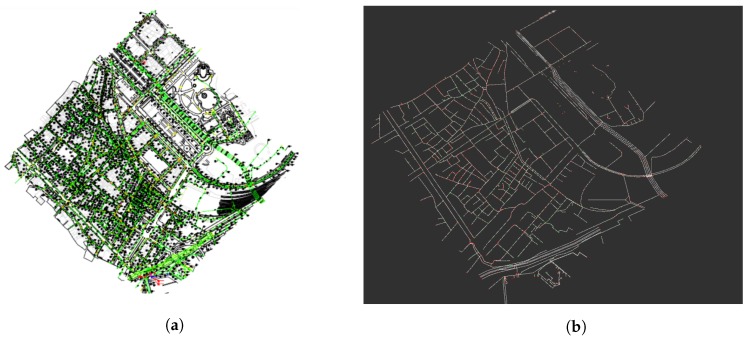
(**a**) The complete GIS information as given by the local authority. (**b**) The extracted graph used by the proposed localization system.

**Figure 3 sensors-19-04946-f003:**
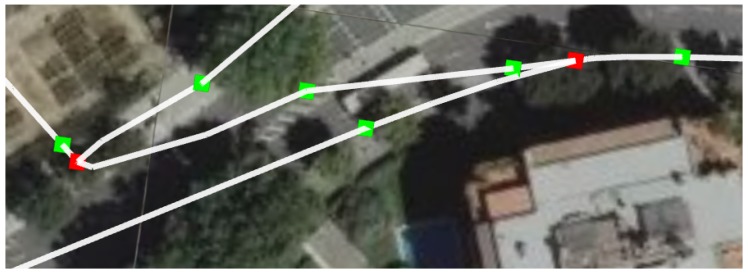
Detail of a sewer graph as used by our proposed localization method. The sewer galleries are marked in white lines. The manholes and forks are marked in green and red squares, respectively.

**Figure 4 sensors-19-04946-f004:**
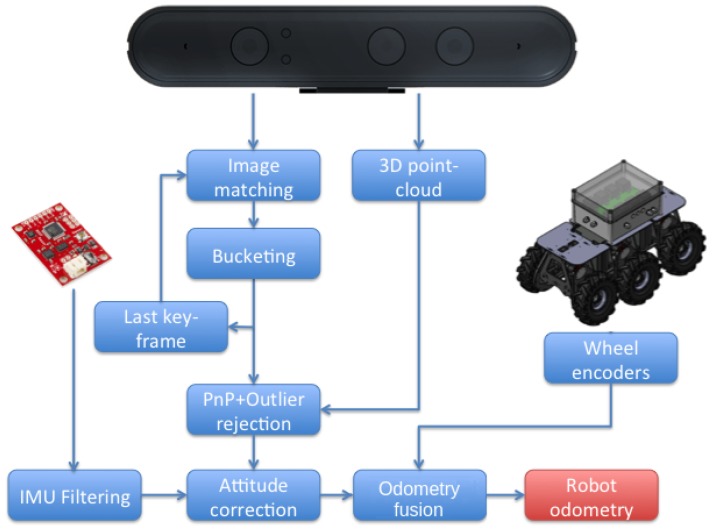
Different odometry sources, including the proposed fused odometry system.

**Figure 5 sensors-19-04946-f005:**
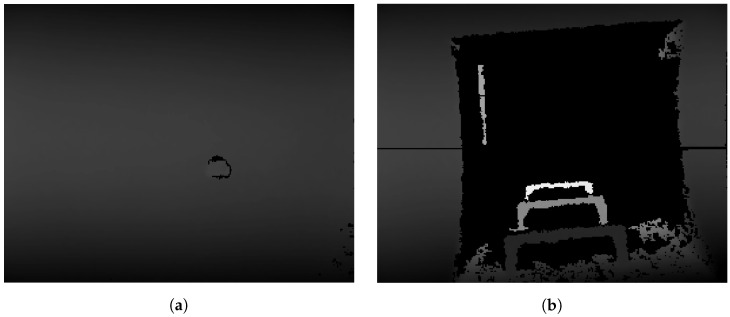
View of the sewer gallery from the depth camera pointing towards the ceiling. (**a**) Regular gallery ceiling. (**b**) Manhole.

**Figure 6 sensors-19-04946-f006:**
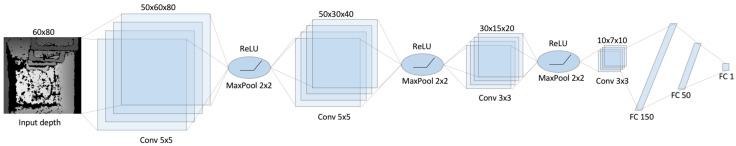
Proposed Convolutional Neural Network for automatic manhole detection.

**Figure 7 sensors-19-04946-f007:**
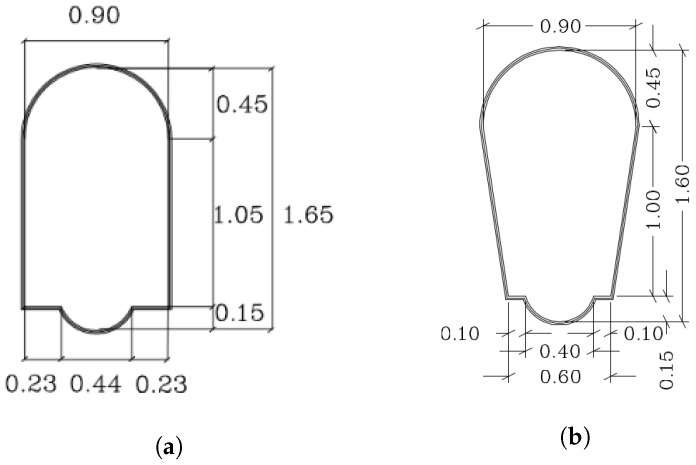
Two typical cross sections of visitable sewers in Barcelona. (**a**) Section of type T130. (**b**) Section of type T111.

**Figure 8 sensors-19-04946-f008:**
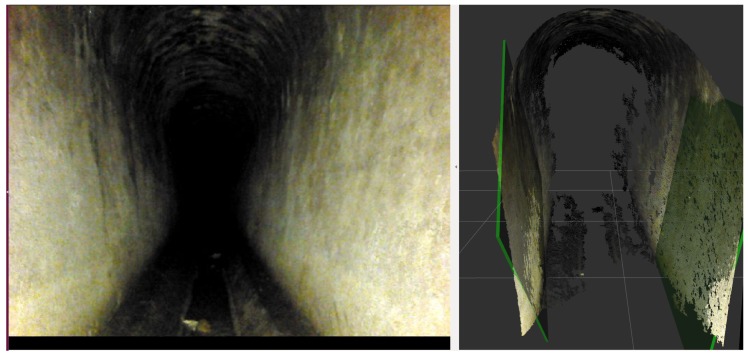
**Left**: Typical section inside the sewers. **Right**: Detected wall planes marked in green.

**Figure 9 sensors-19-04946-f009:**
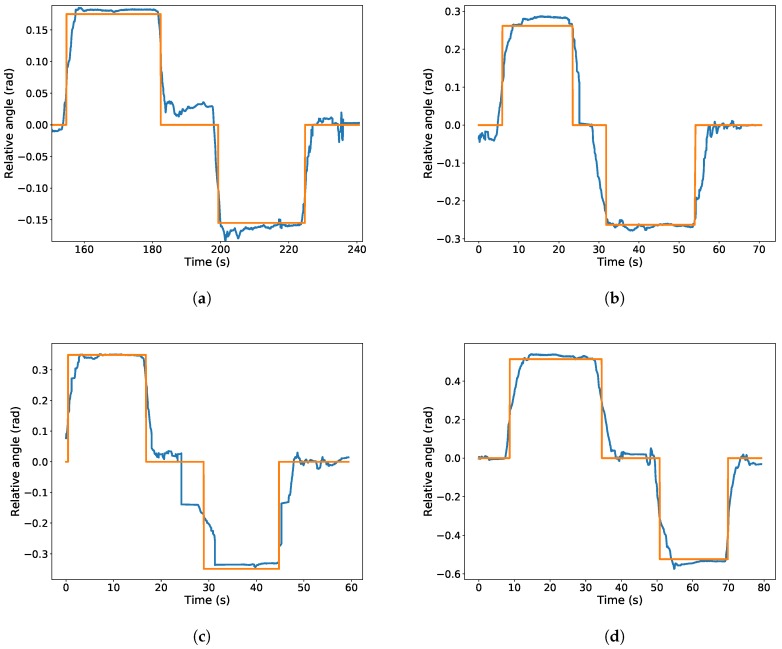
Experimental tests for the relative orientation estimator. The plots represent the estimated orientation in blue and the reference value in orange. (**a**–**d**) The camera is moved to 10, 15, 20 and 30 degrees in each direction, respectively.

**Figure 10 sensors-19-04946-f010:**
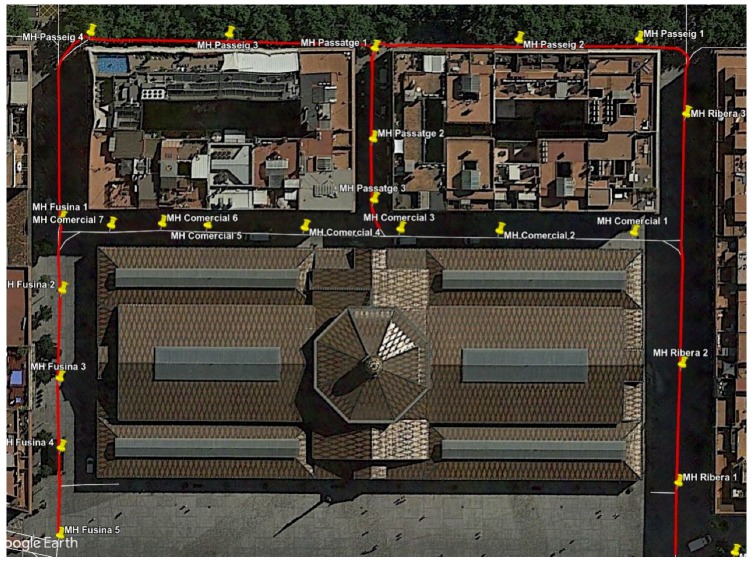
Localization results in the Mercat del Born Scenario overlaid on a satellite view in the experiments of 2017 October, 11th.

**Figure 11 sensors-19-04946-f011:**
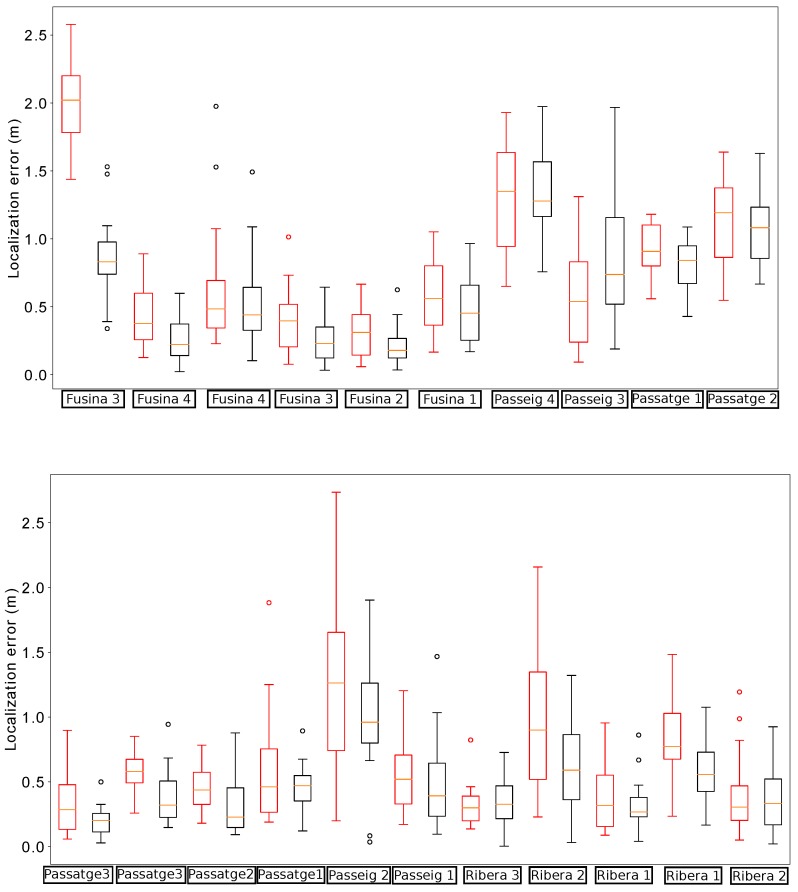
Distribution of errors when passing below the manholes in the localization experiments for testing the impact of the orientation estimation. In each manhole (denoted by the street name and a number) we present two results: the leftmost (in red) is obtained without using the estimations, while the rightmost (in black) uses them.

**Figure 12 sensors-19-04946-f012:**
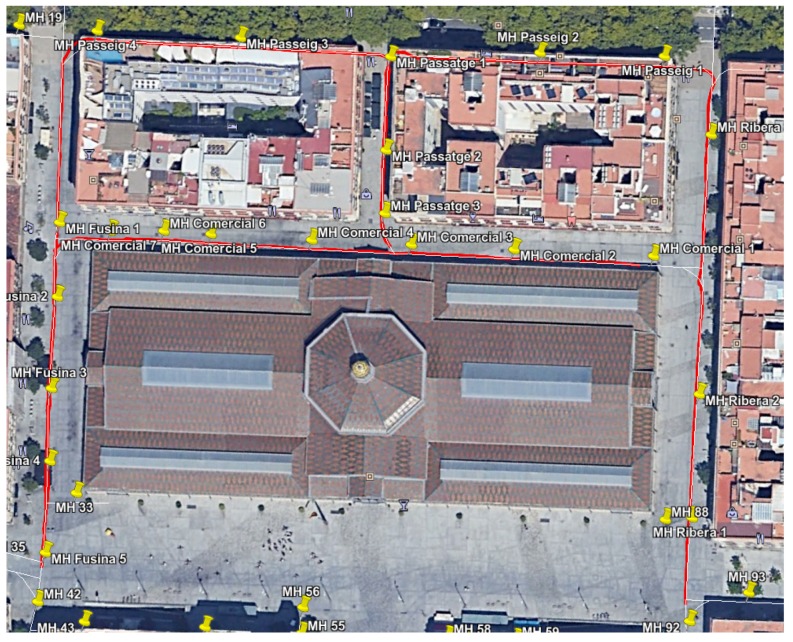
Localization results in the Mercat del Born Scenario overlaid on a satellite view in the experiments of 2017 October, 17th.

**Figure 13 sensors-19-04946-f013:**
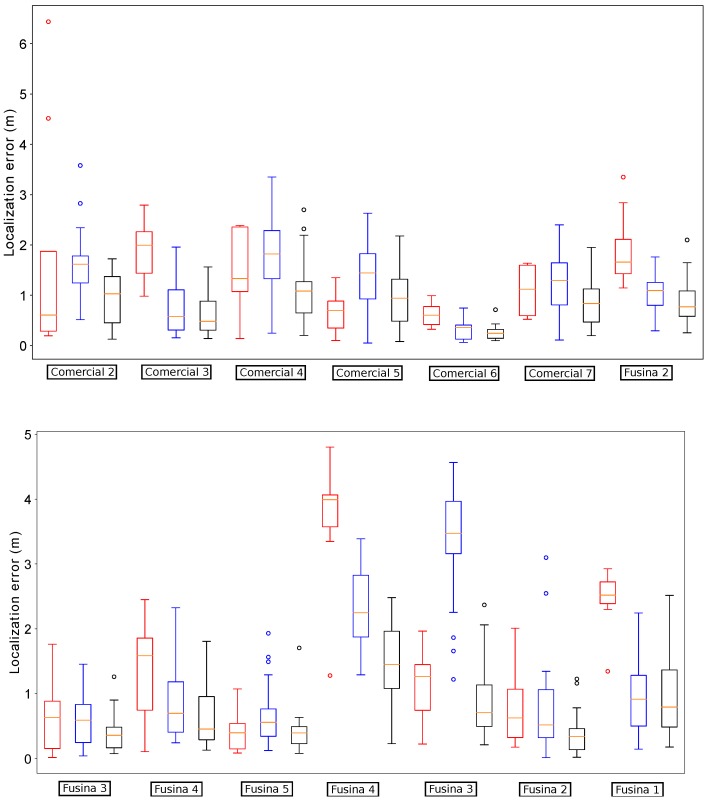
Distribution of errors when passing below the starting fourteen manholes in the localization experiments for odometry comparison. For each manhole we present three measures obtained by using different odometry estimations: VO (leftmost, red), wheel odometry (middle, blue) and fused odometry (rightmost, black).

**Figure 14 sensors-19-04946-f014:**
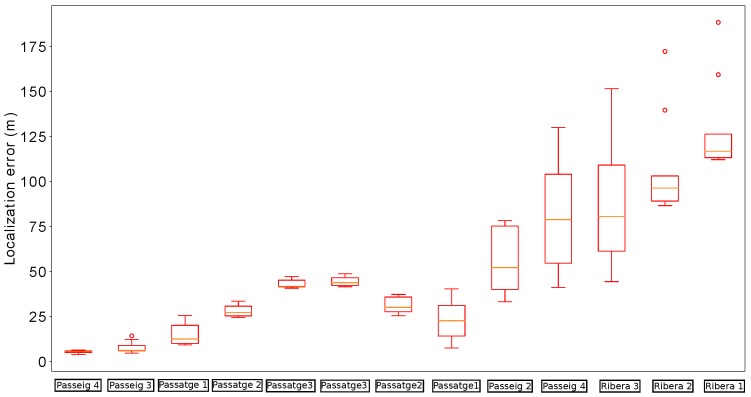
Distribution of errors when passing the remaining thirteen manholes with VO. In this case, the localization system lost the track of the robot.

**Figure 15 sensors-19-04946-f015:**
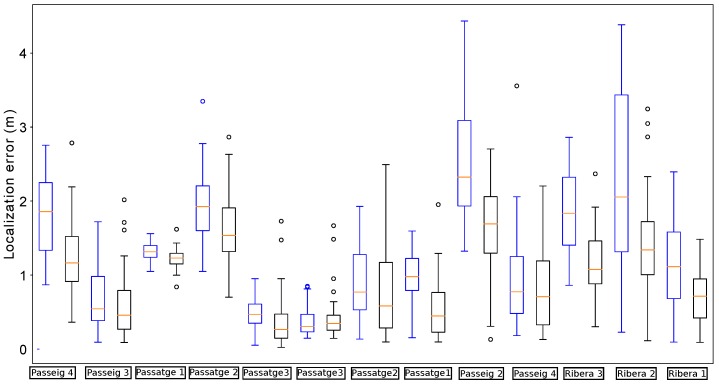
Distribution of errors when passing the remaining thirteen manholes with wheel odometry (leftmost measures in blue) and fused odometry (rightmost measures in black).

**Figure 16 sensors-19-04946-f016:**
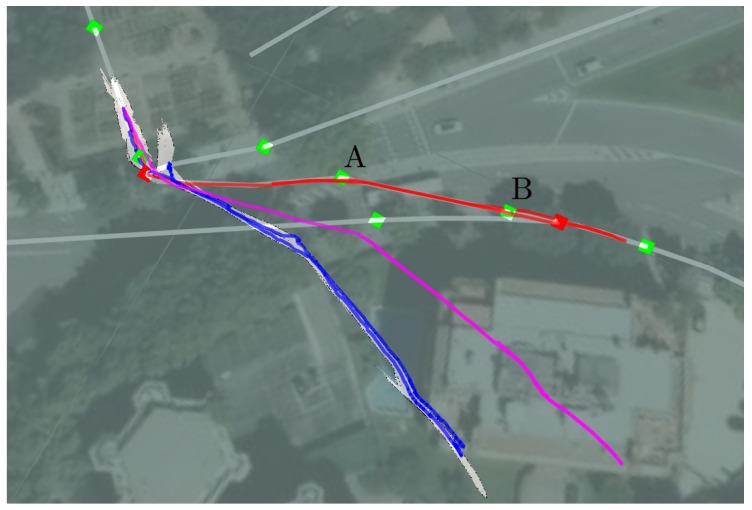
Three examples of localization results are plotted as obtained by our method, gmapping and ORBSLAM-2 in red, blue and purple, respectively. The output map generated by the gmapping method is also represented. The manholes labeled A and B are the ones visited during the experiment.

**Figure 17 sensors-19-04946-f017:**
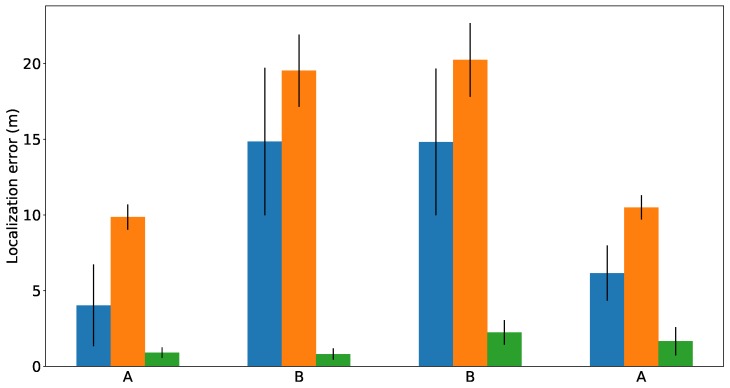
Localization errors obtained by the three methods in Creu de Pedralbes scenario. The results of gmapping, ORBSLAM 2 and the proposed method are plotted in blue, orange and green bars, respectively.

**Figure 18 sensors-19-04946-f018:**
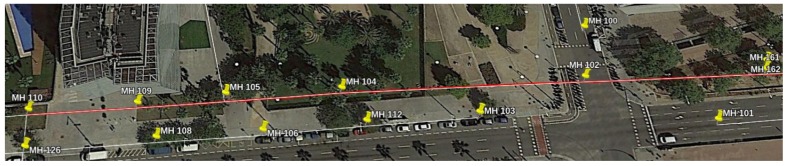
Localization results of the robot’s position in the experiments of Passeig Garcia Faria, on December, 13th.

**Figure 19 sensors-19-04946-f019:**
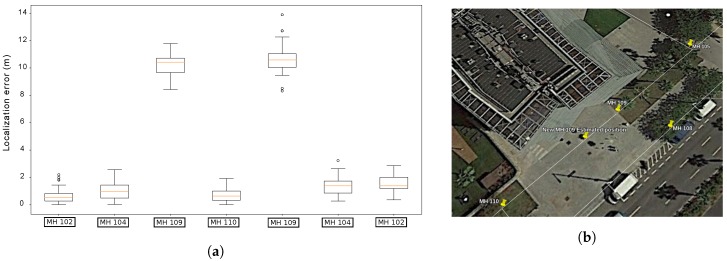
(**a**) Distribution of the localization errors whenever the robot passed below a Manhole in Passeig Garcia Faria. (**b**) Actual location of MH 109, as estimated by our localization system.

**Figure 20 sensors-19-04946-f020:**
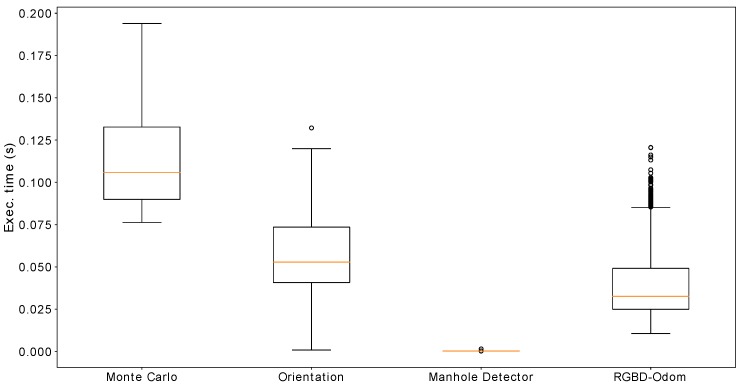
Distribution of the execution times of each process of the proposed localization system.

**Table 1 sensors-19-04946-t001:** Confusion matrix using the proposed approach.

		Predicted	
		Positive	Negative
Actual	Positive	0.95	0.05
	Negative	0.01	0.99

**Table 2 sensors-19-04946-t002:** Confusion matrix using the approach presented in [2].

		Predicted	
		Positive	Negative
Actual	Positive	0.90	0.1
	Negative	0.02	0.98

**Table 3 sensors-19-04946-t003:** Execution time of the main processes of our proposed system.

	Monte Carlo	Orientation	Manhole Det.	RGB-D Odom.
Unit Execution time (ms)	113 ± 27.4	57.5 ± 20.0	0.226 ± 0.042	37.8 ± 16.3
Times called	1223	24,649	24,723	49,915
Global execution time (s)	138.4	1417.9	11.13	930.0

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
