# Peer review of "A Robust Localization System for Inspection Robots in Sewer Networks†"

_sensors, 2019, doi:10.3390/s19224946_

Round 1
Reviewer 1 Report
This paper presents a robust localization system for global pose estimation on sewers. The paper has potential to be further advanced and used in various contexts associated with field robotics. However, this paper has the following problems to be solved.
1. Some sentences have grammar problems. (Such as the sentence on line 86.)
2. The paper evaluates the accuracy of the method proposed, you’d better explain the time efficience of every stage.
3. The related work could be extended and incorporates more comprehensive discussions. Some discussions and important references are missing and should be added in the introduction. The following references maybe helpful to you.
Schmuck, Patrik, and Margarita Chli. "CCM‐SLAM: Robust and efficient centralized collaborative monocular simultaneous localization and mapping for robotic teams." Journal of Field Robotics 36.4 (2019): 763-781.
Mascarich, Frank, et al. "Radiation source localization in gps-denied environments using aerial robots." 2018 IEEE International Conference on Robotics and Automation (ICRA). IEEE, 2018.
Liu, L. , Wang, S. , Hu, B. , Qiong, Q. , & Rosenblum, D. S. (2018). Learning structures of interval-based bayesian networks in probabilistic generative model for human complex activity recognition. Pattern Recognition, 81.
Author Response
We would like to thank the reviewer for his valuable comments and for helping us to make our contribution more valuable. The replies to each point can be found below.
1. Some sentences have grammar problems. (Such as the sentence on line 86.)
We have reviewed the whole text fixing some grammar problems as the one pointed out by the reviewer. All the changes in the text are marked in red for make it easier to track them.
2. The paper evaluates the accuracy of the method proposed, you’d better explain the time efficience of every stage.
We have included an analysis of the computational load of each subsystem that has been added in the new subsection 9.5 of Section 9: Experiments.
3. The related work could be extended and incorporates more comprehensive discussions. Some discussions and important references are missing and should be added in the introduction. The following references maybe helpful to you.
We would like to thank the reviewer for pointing out to-date and relevant references. We used the recommendations for updating the Section 2: Related work. In particular, a new paragraph was added. It can be found in lines 127-140.
Reviewer 2 Report
Overall nice work to show the efficiency of CNN for modeling sewage systems and using robots for sewage maintenance. Experimental evaluation of localization is done. Results are encouraging Only one question remains about the space (storage) requirements and time complexity of CNN. It would be good if the authors can provide some analysis or some discussion on these issues.
Author Response
We would like to thank the author for such an encouraging review.We have done our best to include the requested improvements. All the changes in this revision have been marked in red for making them easy to track. The most relevant changes are as follows:
We have added a new section (9.5) that includes information not only about the time complexity of the CNN, but also of the computational requirements of the main processes of the localization system proposed in this paper.
Regarding to the storage requirement of the CNN, its number of parameters is around 110000, therefore the network is quite small when compared to other modern CNNs architecture in which the number of parameters is usually of several millions. (https://www.jeremyjordan.me/convnet-architectures/).
Last, we have introduced a noticeable amount of changes in order to improve the English language and style.
Round 2
Reviewer 1 Report
This paper deals with some issues about simultaneous localization and mapping. More specifically, the authors propose a robust localization system for global pose estimation on sewers. The topic under study is quite interesting and has attracted interest during the last years. The paper is, in general, well written and easy to follow. The authors present a review of the state of the art and make clear their contribution to the prior works. However, there are some issues that need to be addressed:
1.This paper extends and improves the CNN architecture presented by Alejo et al, it will be better if you compare the method you propose with theirs.
2. This paper presents a method to detect manhole based on CNN. For high dimensional data, dimensionality reduction helps to decrease the computational requirements of the classifier. The related work could be extended, focusing on feature extraction. The following references maybe helpful to you.
Liu, L. , Wang, S. , Hu, B. , Qiong, Q. , & Rosenblum, D. S. . (2018). Learning structures of interval-based Bayesian networks in probabilistic generative model for human complex activity recognition. Pattern Recognition, 81.
Garcia-Gasulla, D., Parés, F., Vilalta, A., Moreno, J., Ayguadé, E., Labarta, J., & Suzumura, T. (2018). On the behavior of convolutional nets for feature extraction. Journal of Artificial Intelligence Research, 61, 563-592.
Wang, L., Wu, J., Huang, S. L., Zheng, L., Xu, X., Zhang, L., & Huang, J. (2019, July). An efficient approach to informative feature extraction from multimodal data. In Proceedings of the AAAI Conference on Artificial Intelligence (Vol. 33, pp. 5281-5288).
Author Response
This paper deals with some issues about simultaneous localization and mapping. More specifically, the authors propose a robust localization system for global pose estimation on sewers. The topic under study is quite interesting and has attracted interest during the last years. The paper is, in general, well written and easy to follow. The authors present a review of the state of the art and make clear their contribution to the prior works. However, there are some issues that need to be addressed:
>Dear reviewer, thanks very much for your time, we really appreciate your contributions to improve this paper. You can find the changes in the second revision marked in blue in the document.
1.This paper extends and improves the CNN architecture presented by Alejo et al, it will be better if you compare the method you propose with theirs.
> For the sake of clarity, we included the confusion matrix of the classifier presented in Alejo et al., and we compared the results of both classifiers. This has been included in the reviewed document in the paragraph starting at line 456.
2. This paper presents a method to detect manhole based on CNN. For high dimensional data, dimensionality reduction helps to decrease the computational requirements of the classifier. The related work could be extended, focusing on feature extraction. The following references maybe helpful to you.
Liu, L. , Wang, S. , Hu, B. , Qiong, Q. , & Rosenblum, D. S. . (2018). Learning structures of interval-based Bayesian networks in probabilistic generative model for human complex activity recognition. Pattern Recognition, 81.
Garcia-Gasulla, D., Parés, F., Vilalta, A., Moreno, J., Ayguadé, E., Labarta, J., ... & Suzumura, T. (2018). On the behavior of convolutional nets for feature extraction. Journal of Artificial Intelligence Research, 61, 563-592.
Wang, L., Wu, J., Huang, S. L., Zheng, L., Xu, X., Zhang, L., & Huang, J. (2019, July). An efficient approach to informative feature extraction from multimodal data. In Proceedings of the AAAI Conference on Artificial Intelligence (Vol. 33, pp. 5281-5288).
> Thanks very much for the bibliography suggestions. We included them into the state of the art and the CNN sections (sections 2 and 7.1). In particular, we changed the paragraph starting at line 404 of section 7.1. We also took the chance to highlight in the document that this approach does not need dimensionality reduction due to image down-sampling performed prior to classification (lines 412 and 413). Finally, we highlight some relevant machine learning applications in the state of the art section (lines 147-150).